# The Effect of Geographical Indications (GIs) on the Koerintji Cinnamon Sales Price and Information of Origin

**Sidi Rana Menggala** [1,*], **Wouter Vanhove** [1], **Dimas Rahadian Aji Muhammad** [2], **Abdur Rahman** [3], **Stijn Speelman** [4] **and Patrick Van Damme** [1,5]

1   Laboratory of Tropical and Subtropical Agriculture and Ethnobotany, Faculty of Bioscience Engineering, Ghent University, Coupure Links 653, 9000 Gent, Belgium; wouter.vanhove@ugent.be (W.V.); patrick.vandamme@ugent.be (P.V.D.)

2   Department of Food Science and Technology, Sebelas Maret University, Surakarta 57126, Indonesia; dimasrahadian@staff.uns.ac.id

3   Rahman Kerinci Cinnamon Community, Pasar Senen Village, Siulak Sub-District, Kerinci District 31760, Indonesia; abdurrahman1310@gmail.com

4   Department of Agricultural Economics, Faculty of Bioscience Engineering, Ghent University, Coupure Links 653, 9000 Gent, Belgium; stijn.speelman@ugent.be

5   Faculty of Tropical AgriSciences, Czech University of Life Sciences Prague, 165 21 Prague, Czech Republic

\*   Correspondence: sidirana.menggalasusanto@ugent.be

**Abstract:** This paper focuses on the impact of the geographical indications (GIs) of Koerintji cinnamon on its value chain. The study was performed from September 2017 to November 2017 in Talang Kemuning, Kerinci regency, Indonesia. A village farmers' group was surveyed using a semi-structured questionnaire, roundtable discussions, interviews, and direct observations to understand whether the GIs improve income, source of production, and promote product quality. Using a descriptive method, the literature on the topic was analyzed, and a value chain study was structured from the review's findings. This helped us to better understand how GIs' effects are dispersed among the chain actors and eventually enter the areas from which GI commodities originate. GIs generate added value, especially for farmers and buyers. Specifically, by using the GI Koerintji cinnamon's handbook of requirements, the efficiency of farmers and buyers has improved. The studied population included farmers from Koerintji Cinnamon Jambi Geographical Indication Protection Society (MPIG-K2J) and Tani Saktik Alam Kerinci (TAKTIK), a farmer group. To obtain a GI, TAKTIK had to implement good agriculture practices and good manufacturing practices based on the handbook. Results show that farmers developed a system to monitor post-harvest handling in assuring a safe and high-quality supply to the global spices market. Following the book requirements, TAKTIK members follow specific procedures, including selecting raw materials, grading, origin verification, and quality control. Furthermore, GIs enable farmers to claim a price premium. As a result, GI Koerintji cinnamon's presence has added value and credibility to TAKTIK farmers, leading to price improvement.

**Keywords:** geographical indications; value chain

## 1. Introduction

Indonesia is the world's leading producer and exporter of cinnamon (*Cinnamomum burmannii* Blume), accounting for 40% of the total global spices market. For hundreds of years now, the country has been part of cinnamon's history [1], an evergreen tree that grows well depending on several factors, including altitude, rainfall, soil condition, topography, and groundwater availability [2]. Various parts of the cinnamon tree can be used, including the root, trunk, leaves, and bark. However, the most valued part is the bark, the outer layer that covers the trunk. The bark is traditionally peeled, sundried, and differentiated into grades based on tree age. Cinnamon is commonly used as a flavoring additive and an aromatic condiment in the food and beverage industry, mainly to add flavor to drinks and

cuisine. Cinnamon has a characteristic scent and is widespread as a drink and exotic food mixture among buyers and consumers [3].

Cinnamon from the Kerinci regency in Sumatera is the most renowned cinnamon brand in the global spice market [4]. The Koerintji cinnamon, known as a premium cinnamon commodity in the spice market, is the high volatile oil in its bark. Commonly, international buyers seek this specific oil to blend it with other spices and herbs for derivatives goods. In the past few years, Koerintji cinnamon has recently been used to manufacture various products, including chocolate and cocoa drinks, to improve their taste [5].

As the popularity grew, there was a noticeable impact on the value chain process. The market demands a premium commodity of cinnamon from this center of cinnamon production in Indonesia. Here, cinnamon smallholder farmers play an essential role in the cinnamon value chain, with most of the production coming from forestland of less than one to two hectares [6]. In 2017, there were approximately 11,000 smallholder cinnamon farmers in Kerinci whose income largely depends on this forestry product. The forestland planted with cinnamon trees is spread over seven mountainous districts of the Kerinci Regency with a total surface area of 40,762 ha [7]. According to the Kerinci Agriculture Agency, the production could reach 53,385 metric tons of dried bark in 2017, with most of the volume being sold to international markets [1,7]. To prove this point, the exported cinnamon exported is compared with reports export statistics from Indonesia to the global market [6]. According to the latter resource, cinnamons' global market was estimated at 230,000 metric tons in 2017 and is expected to reach 250,000 metric tons by 2024, including raw material that was coming from Indonesia. In 2017, Koerintji cinnamon had a global market share of 40%, with a trade value of $148,075,756 [8].

Since 2013, more international buyers source cinnamon directly from the Kerinci regency, hoping to deal with the farmers directly. However, the increasing demand could not match production quantities, causing some middlemen to source the bark from other locations [9]. This became a problem for the traceability aspects. Another concerning issue is related to the growing demand for Koerintji cinnamon, in which some processing companies blend the Koerintji cinnamon bark products with lower-quality cinnamon barks from other areas. The premium volatile oil that are average 4.5 per cent coming Kerinci regency that were mixed with lower quality of 2–3% from other regions which are not favorable for certain processing companies using this spice as a mixture and need a certain amount of volatile oil percentage.

For this reason, local and international buyers questioned the originality of Koerintji cinnamon. Apart from mixing with non-Koerintji cinnamon, the quality of the original Koerintji cinnamon itself has become a significant problem. The introductory part of this paper showed that an urge towards more product quality and originality is perusal in both the global spice market and to the consumer. Some literature also refers to GIs as an identity that indicates an item's origin and a product due to geographical, environmental, natural, and human factors which give reputation, quality, and specific characteristics to products and goods [10,11].

GIs may protect a product origin (*sui generis*) and improve their processing activities to meet good agriculture practices [12]. More than 10,000 Geographical Indications (GI) exist around the world. They aim at "identifying a good as originating in the territory of a State, or an area or locality within that territory, where a specific quality, reputation, or other characteristics of the commodity is, fundamentally, attributable to its geographical origin" (WTO definition). When producers obtain a GI for a specific commodity, they get the exclusive right to name their products after the territory where the commodity is produced. Therefore, GIs can be considered as a tool for adding value to certain products. GIs act as a quality assurance that unlocks value by capitalizing on consumer's desire for diverse, typical, quality products [13]. GIs can ensure compliance with the origin, quality, safety, and premium price. As a recognizable label, it promotes a product among different actors in the value chain, buyers, and consumers [14].

The present paper aims to identify the effects of GIs on the Koerintji cinnamon sales price increase in a farmers' organization, named TAKTIK (Tani Sakti Alam Kerinci), as the first executor of this label in Kerinci regency. TAKTIK also represents farmers and acts as a link between them and market players. With the assistance and guidance by the GI Korintji label holder which is the Masyarakat Perlindungan Indikasi Geografis Kerinci (MPIG-K2). The GI Koerintji Kayumanis certificate was then published on 26 May 2016 by the Ministry of Law and Human Rights (ID G000000043), see in Appendix A. The Koerintji cinnamon was the 43rd product-protected geographical indication registered in Indonesia [15]. Koerintji is the old spelling of Kerinci regency, while *Kayu Manis* is the word in the Indonesian language for cinnamon [16]. After the GIs was registered, MPIG-K2J was responsible for managing them.

MPIG-K2J emphasizes GIs protection of Koerintji cinnamon in order that productivity improvement that farmers should follow to meet the GI requirements. The initiative to register Koerintji cinnamon as a GI label by MPIG-K2J was an effort to add value to the commodity. TAKTIK was a pilot project for MPIG-K2J to improve their production activities to receive higher sell prices in the market by following the Buku Persyaratan Indikasi Geografis Kayumanis Koerintji Jambi (Handbook of Requirements for the Geographical Indications for Koerintji Cinnamon).

## 2. Theoretical Background

### 2.1. Geographical Indications in the Value Chain

A value chain is a range of activities required to bring a product from production to the final consumer [17]. In order to address the value chains, GI was identified as a possible adequate sign of sustainability and origin to be developed to strengthen the capabilities of producers and private enterprises, and effectively link cinnamon farmers to domestic and international markets through the development of GI value chains. This is expected to increase smallholder producers and agribusinesses' increased income in the prioritized GI value chain.

According to Fernandez-Stark et al., traceability, administrative measures, and the product origin can become an upgrading trajectory [18]. In analyzing a cinnamon value chain system, the main activities and supporters were investigated. Based on the concept of Porter [19], a chain system has two activities, namely the main activity and the supporting ones. Another literature shows that the farmers have a limited understanding of upgrading opportunities regarding the production methods. In this case, the stakeholder who knows this certification of origin sees it as a marketing purpose but not as guidance for process upgrading including biodiversity concerns, sustainable agriculture practices, and food safety [20]. Until recently, GI registration's social-economic and environmental impacts are still debatable for researchers' implementation and require substantial investment and time [21].

A GI-based product can generate a premium brand price and contribute to local employment, ultimately helping local community development [22]. Therefore, upgrading cinnamon farmers' production by adopting geographical indications can be a part of a broader agenda in the international development practice referred to as 'value chain development (VCD) [23,24]. Recognition of a GI affects the local value chain's structure and strength [25–27]. Unfortunately, most registered GIs in Indonesian agriculture are marketed merely as a commodity without strategically maximizing the emphasis on their GIs in the global market [28]. According to Menapace and Moschini [29], GI certification improves the ability of companies to leverage reputation to assure consumers of product quality.

### 2.2. Geographical Indications as Value-Added

Since the 21st century, there has been an increasing demand for local, traditional, and more extensively produced food [30]. Products with GI labels can contribute to these consumption trends. One of the essential aspects of GI is that it can increase the economic viability of smallholder enterprises from whom such food often sources. However, series

of studies show that different GIs impact price premiums [31,32]. In the literature review, some scholars also mentioned that GI is a tool for agricultural and cultural development in developing countries, strengthening local products' value proposition [33,34]. The researcher here wants to focus on the value proposition that can become a rural development tool, serving commercial and economic interests while preserving local values [25,35] and also having multiple aims such as: (1) to increase the selling price of a product in a market of growing competition; (2) to give a piece of controlled information about the origin to the consumer, at a time where value chains are less and less traceable; and (3) to promote local and rural development by boosting collective dynamics directed to produce high economic value.

Add to this, according to Ramli [36], the application and implementation of GIs, particularly in Indonesia, has various benefits including: (1) legal protection from the Indonesian Ministry of Human Rights and Law as intellectual property rights; (2) marketing assets for domestic and foreign trade; (3) improving the value-added of the product improving the regional economy; (4) improving product reputation in global trade; (5) providing means to avoid false competition in the way of belonging to certain producers in the geographical area defined who comply with the specific conditions of production for the product.

### 2.3. Geographical Indications as a Label of Origin

GIs link the product's origin with more specific functionality, which is safer and more hygienic. For example, consumers in Europe are concerned about the quality of their food, which can be more guaranteed when they can trace the products back to their origin. With this terminology, GIs may change the status of an agricultural product from "commodity" to "origin product" [37]. This shows that consumers respond positively to a specific label, even when unaware of the specificity associated with the indicated geographical origin [38]. When appropriately used and well-controlled, the label can become a powerful marketing tool and contribute to a region's economic development [39]. Giovannucci et al. [31] state that GIs are a tool for "institutionalizing the resources of a place." The label may also differentiate products based on unique local features, historical-cultural factors, and characteristics linked to natural, geographical, and human factors such as soil, climate, local know-how, and traditions that integrated origin-labeling [40]. This label may protect certain goods from specific geographical locations from misuse and imitation, promoting rural development and help consumers by giving them information concerning their specific characteristics [41].

### 2.4. Research Hypothesis

We hypothesize that a GI is an essential development tool for cinnamon farmers that impacts sales prices, quality, and information for consumers on the source of origin. Moreover, The Indonesian Ministry of Law and Human Rights stated that GIs are powerful devices that enhance value and rural economic development, in line with the nation's economy [35]. Therefore, the research shall identify whether the GI label can impact the cinnamon sales price and increase farmers' revenue. Furthermore, going deeper into the research will also reveal the impact of the value chain among the stakeholder. Thus, the GI of Korintji cinnamon can also address the issues of sale price, source of origin, and how value can be created, transferred, and distributed along the supply chain [31,42]. In the following Figure 1, we see this situation illustrated graphically.

The traditional supply chain is shown from point one to five, from farm to fork. Starting at point six, the hypothesis begins where global consumers need to be clarified of product origin. Consumers demand transparency to the wholesaler and supermarket at point seven. Wholesalers work together with an agricultural support agency for geographical indications to improve their production and manufacturing. The agency provides training and guidance for GIs so that, later, the label can be put on their packaging. The farmers' group two are chosen as the partner and will register the GI as a legal provision for the brand of origin. The farmers' group also improves its traceability system so that all

transactions can be traced to its members. The farmers' group, empowered with a label of origin, improvised production, and accounted traceability, herewith have a better farm gate price to offer to the buyer that is mostly processing and has a license to export. The reasons are that commonly this processing company mix and blends all the bark together. Hence, the GI label present in the newest model of the value chain; processing companies will consider twice whether to blend products due to the traceability system that has been created before. From point nine to twelve, the importer can recognize the cinnamon bark from the village and producer through the supply chain development and the GI mechanism.

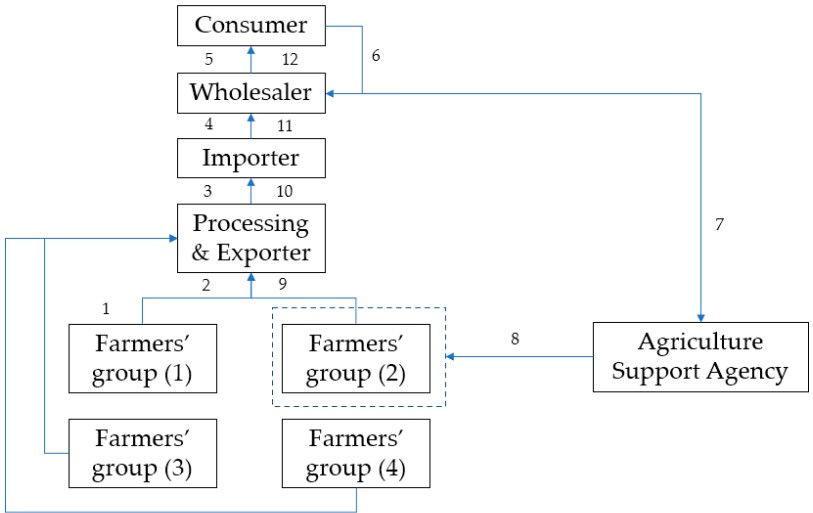

**Figure 1.** The research hypothesis. Source: Researcher, 2017.

## 3. Methodology

### 3.1. Research Framework

The research combined questionnaire surveys, interviews, and round table discussions to collect cinnamon harvesters' information and data. The collected data would help develop the best practice framework and the conclusions and recommendations for findings [43]. In addition, the research findings aim as a vehicle for communication and negotiation among the stakeholders and a cornerstone of collective action. Its implementation is critical to the process's success and long-term viability.

Figure 2 shows the conceptual research framework. GIs' current practice literature will be reviewed, including books, journals, proceedings from conferences, workshops, seminars, and other sources. Past and present methods of GIs will also be investigated during the research. The review exercise includes developing an instrument to conduct the interviews and questionnaires [37]. Finally, the information collected from these interviews and questionnaire surveys is consolidated to answer the research objectives.

### 3.2. Research Approach

This study uses a descriptive approach, beginning with GIs' research framework and value chain, a common sense for linkage as added value-based. It starts with a research question, "What is The Effect of Geographical Indications (GIs) on the Koerintji Cinnamon Value Chain for Greater Farmers' Income?" The hypotheses is answered by collecting empirical data in the field visits and conducting questionnaires and round table discussions. The research focus was on answering whether GIs can play a special role in improving farm practices and income.

### 3.3. Data Collection

In this case study research, multiple data were obtained from multiple sources, such as ISI, Web of Knowledge, Science Direct, and Google Scholar. The initial search syntax used was "geographical indications", in combination with "value chain", "added

value", "roles of NGOs", "forest-commodities", and "cinnamon". The field observation occurred in September 2017, including interviews, observations (direct and participant), questionnaires, and relevant documents. There are two methods of identifying the research objectives, which are used in the research. They are stakeholder perception and business model canvas.

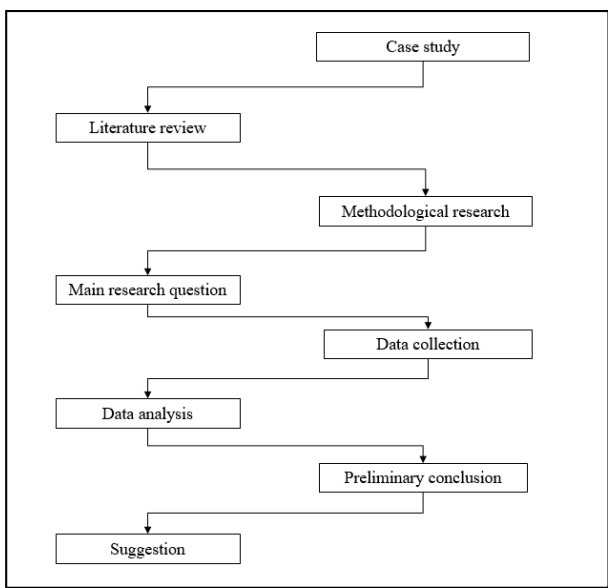

**Figure 2.** The conceptual research framework.

The reasoning of stakeholder perception is to analyze the importance of stakeholders' emotions, experiences, and perceptions of consequences. Emotions are senses that stakeholders feel during an experience. They are momentary and relate to fairness, trust, esteem, dignity, etc. These dimensions are related to stakeholders' perceptions of the service delivery [44,45].

In order to in-depth research, a business model canvass was used to understand this sector's process to analyze the increasing competitiveness of the label [46].

*3.4. Respondents*

Before in-depth interviews, a round table discussion with various stakeholders of the cinnamon industry, from producers to local government, was conducted on 19 October 2017 with 18 respondents. A round table is an academic discussion where participants agree on a specific topic to discuss and debate. Each person was given an equal right to participate. The discussion was aimed to identify the constraints and opportunities of the cinnamon industry. The participants of RDs were invited purposively from a list provided by Rikolto. An in-depth analysis was used to identify the bottlenecks, interventions, and other collective ideas in the cinnamon sector. Collectively, the discussion could gain farmers' insight and experiences and uncover their role as key actors at the local level.

The research used semi-structured interviews to collect information from farmers and other relevant actors, such as the village head, buyers, and NGO workers. Each area was snowballed for the samples encountered on the road based on the targeted 20 individuals ($n = 20$) members of TAKTIK in a different area, a total of 60 members ($n = 60$). In addition, the socio-demographic and characteristics of the three areas were analyzed that can be seen in Table 1.

**Table 1.** Socio-demographic characteristics of the targeted farmers' group of TAKTIK members.

| Description | | N = 60 | | |
|---|---|---|---|---|
| | | Area A (*N* = 20) | Area B (*N* = 20) | Area C (*N* = 20) |
| Gender | Male | 12 | 14 | 18 |
| | Female | 8 | 6 | 2 |
| Age | Up to 30 | 4 | 4 | 2 |
| | 30–40 | 10 | 8 | 9 |
| | 41–50 | 5 | 5 | 7 |
| | More than 50 | 1 | 3 | 2 |
| Education | Basic school | 10 | 13 | 16 |
| | Secondary school | 6 | 5 | 3 |
| | High school | 4 | 3 | 1 |
| Harvest experience (years) | Up to 10 | 4 | 2 | 2 |
| | 11–21 | 8 | 9 | 10 |
| | 22–31 | 6 | 4 | 5 |
| | More than 31 | 2 | 5 | 3 |
| Forestland size | Small scale (1 ha) | 6 | 7 | 5 |
| | Middle scale (2–3 ha) | 9 | 8 | 12 |
| | Large Scale (3–5 ha) | 5 | 5 | 3 |

Source: Research data collection.

To carter more insights and evidence, an additional structural questionnaire and in-depth interviews were conducted for collecting primary data from farmers (*n* = 20) and relevant stakeholders of 60 (*n* = 60) in Talang Kemuning district from 20 October onwards. The participants were selected based on their willingness. A sampling technique using a structured questionnaire and in-depth interviews were applied to collect primary data from harvesters and relevant stakeholders [47].

### 3.5. Study Site

Kerinci Regency is one of the regencies in the Province of Jambi, Sumatera. Kerinci Regency is located at 01°40′ and 02°26′ S and 101°08′ to 101°50′ E and has an area of 3328.14 km$^2$. More than half of the total area, or rather 1990.89 km$^2$, is Kerinci Seblat National Park, and the remaining 1337.15 km$^2$ are used for cultivation areas and settlements. The research location was in Talang Kemuning village and surrounding area of Bukit Kerman District, Kerinci Regency, Sumatera, Indonesia. The data collection for the research was conducted in the triangle shape as pointed in Figure 3. The main goal was to collect and derivation of qualitative information about the structural aspect of cinnamon production in this village and surrounding.

Talang Kemuning village's total population amounted to 520 households, with 239 residents living outside the village in 2018. According to the village leader's information, 60 percent of the community were farmers and harvesters. In comparison, 40 percent worked in the nearest city of Sungai Penuh with a non-farming-related activity. The village is in a mountainous area, about 500 m above sea level, and covers 1600 ha, with a hilly and mountainous region. This village is a buffer for the Kerinci Seblat National Park (TNKS) conservation area, which borders the Production Forest of Community Utilization Pattern (HP3M) area [48].

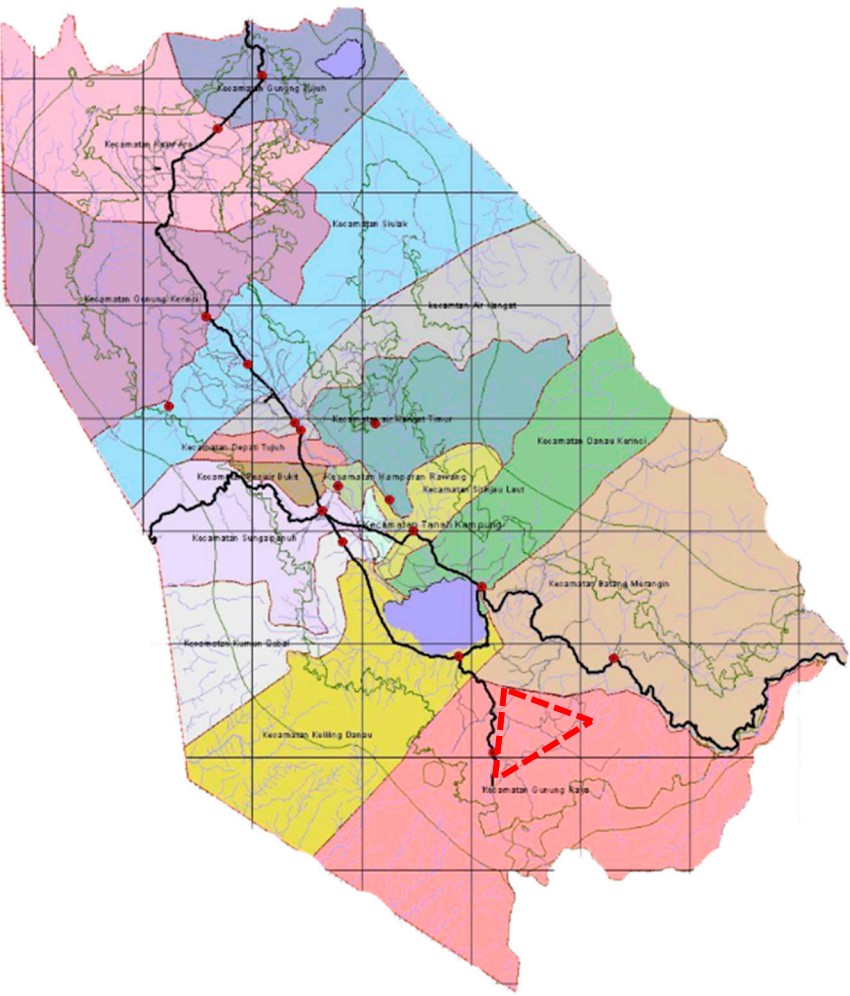

**Figure 3.** Talang Kemuning village and surrounding [triangle], Bukit Kerman sub-district. Source: Kerinci regency regional land agency, 2017.

*3.6. Data Analysis*

The descriptive data analysis proved whether the GIs on the Koerintji Cinnamon affected the value chain for farmers' income. After collection, data were analyzed qualitatively, combining literature review and theoretical basis with field research by interpreting primary and secondary data.

Most of the primary data were from direct observation, round table discussions with 18 participants, and interviews based on their willingness. Specifically, a total of 60 harvesters could be interviewed with questions focusing on the members' perceptions and understanding of GI and its potential using a questioner.

The data was processed with NVivo 12 to evaluate the qualitative data, and a content analysis approach was used. The research started by transcribing the interviews and going over the information. The secondary data were then grouped into categories and coded at the same time as the primary data. For example, stakeholder perceptions, findings of the business model canvas, and members who applied the handbook of requirements were all coded.

The next step was drawing conclusions and verification. After data on GIs were collected and described, research will conclude whether GI brings an impact on the farmgate price and label of origin, which is Kerinci regency.

## 4. Results

### *4.1. Geographical Indications*

According to the leader of MPIG-K2J, GI is a tool for communication and negotiation among the stakeholders along the supply chain. To be able to fulfill these responsibilities, three functions are involved: (1) uniting business actors in the value chain; (2) managing quality, unique characteristics, and product sustainability; and (3) promoting and maintaining product reputation. These responsibilities were also written down in the "Book of Requirements" for Koerintji cinnamon, explained in the following section. MPIG-K2J uses this book to guide farmers willing to participate and use the GI label.

Being a contributor to the cinnamon commodity in Indonesia, MPIG-K2J had encouraged the Sakti Alam Kerinci Farmer Organization (TAKTIK in Talang Kemuning) to innovate continuously. In 2015, this organization also succeeded in legalizing cinnamon patents by achieving the GI certification award, which the Jambi Provincial Plantation Office facilitated. GI certification is a certificate given by the Director-General of Intellectual Property Rights at the Ministry of Law and Human Rights, the Republic of Indonesia, for superior native commodities as an effort to protect the authenticity and uniqueness of agricultural products produced in the "Kerinci regency'. This certification may increase Indonesian agricultural products' competitiveness in domestic and global markets, including cinnamon with its recognizable label (Figure 4).

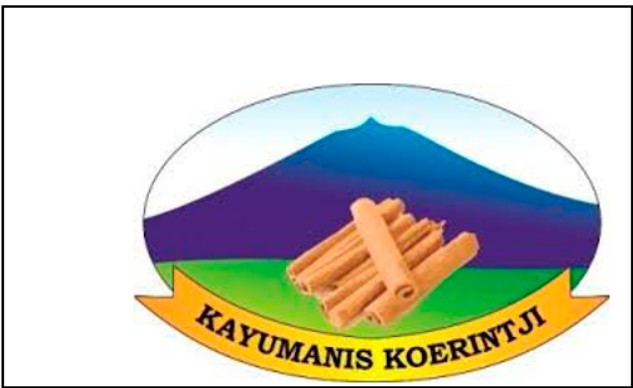

**Figure 4.** The Koerintji Cinnamon GI label. Source: GI Koerintji cinnamon—books of requirement.

With the presence of this label, sellers cannot confess carelessly of selling Koerintji cinnamon. The next step was to ensure that geographical indication recognition is recognized as a commodity of premium quality and has characteristics not similar elsewhere.

### *4.2. GIs Stakeholder Perception*

Understanding stakeholder perception was a necessity to address the research problem. Economic implications are the most important factor related to established geographical indicators, while local employment is essential for stakeholders in the social dimension. In reality, the research needs to access the highest points are given to the price premium, regional value-added (production improvement), and traceability.

With the help of Rikolto (a Belgium non-profit organization based in Indonesia), a roundtable discussion was conducted on the 19th of October 2018 in the Mahkota Hotel, city of Sungai Penuh, Kerinci regency. Various stakeholders were invited to the session, from farmers to local government officials. The event was held to assess whether GIs can become an added value for the value chain. In addition, a questioner was distributed to the stakeholder ($n$ = 18) during the RDs to verify their understanding which can be seen in Figure 5 as follow.

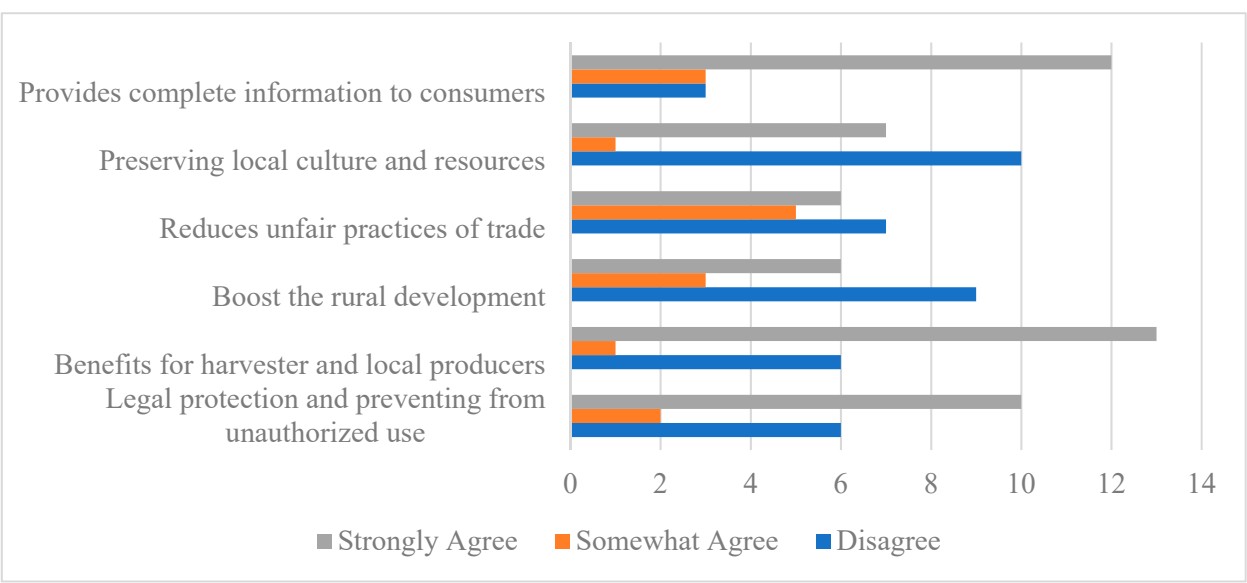

**Figure 5.** The participants' perception for GIs during the round table discussion.

Based on the stakeholder (Figure 5) perception, that *n* = 12 strongly agrees that GI provide information of production origin of the korintjie cinnamon, there were also some disagreements whether enough of the yield was coming from the Bukit Kerman subdistrict. The interesting part, which became a debate during the discussion, was about the benefits of GI. Thirteen people agreed that more buyers are attracted to buy cinnamon with a GI label on its packaging nowadays. It was strongly agreed that GIs benefit them and local producers through legal protection and prevention of any unauthorized use. However, it was vital to determine whether GIs benefit the farmers' production skills and capacities to answer the research question. The participants agree that GI registration will help boost the commodity's competitiveness in both domestic and international markets.

For further facts findings, the researchers used a 'business model canvas' distributed among the participants in Figure 6. Using this canvas is for developing new business models and documenting existing ones in the cinnamon sector. The business model canvas is helpful for the research to identify the most central question such as (1) key partners, (2) key activities, (3) key resources, (4) value propositions, (5) costumer relationships, (6) channels, (7) costumer segments, (8) cost structure, and (9) revenue streams.

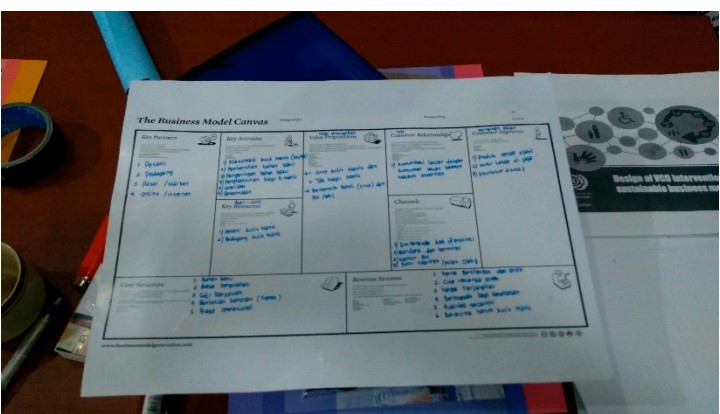

**Figure 6.** Business Model Canvas for GI Koerintji cinnamon.

The business canvas model workshop shows that the participants (*n* = 18) have typical results in Table 2.

**Table 2.** The business model canvas of cinnamon in Kerinci.

| Business Model | The Output of Business Canvas |
|---|---|
| Key Partners | Farmers' contract with partners (ATN & Cassia Coop), Agricultural agencies (Rikolto, MPIG-K2J, KPHP), Kerinci agriculture department, and Jambi provincial agriculture department |
| Key Activities | Operation and production, producing good quality of cinnamon bark, sales of bark, and sales of tree stem and intercrop with seasonal crops |
| Key Resources | 2–5 Ha2 of land, private savings, farmers' organizations' fund, and bank loans |
| Value Propositions | Organic cinnamon, fulfillment of international standards and quality, GI label of Koerintji cinnamon |
| Customer Relationships | High quality of bark (volatile oil), organic cinnamon, and system operational procedures (SOP) |
| Channels | Companies in the city of Padang & Jambi (main channel), sales on-farm, and social media (Facebook group & Whatsapp group) |
| Customer Segments | Processing & exporting companies (ATN, Cassia Coop, Tripper, and CV Kerinci Agro), Middleman & local distributor, Small-medium enterprises |
| Cost Structure | Animal feed (water buffalo), organic fertilizer, gunny bags, land rental, oil for transportation (car & motorcycle), fee for the cultivator, and cleaning of tall grass |
| Revenue Streams | KM (€2.50–€2.80/KG), KF (€2.40–€2.60/KG), KS (€2.20–€2.40/KG), KA (€2.30–€ 2.50/KG) X harvest amount |

Note—revenue streams: KM is bark age to 25 years old, KF = 15–25 years old, KS = 8–14 years old, KA = 5–8 years old.

The research explored more information to answer the research questions from this stakeholder perceptions of GI Koerintji cinnamon and business model canvas. The research continues to validate directly on farms, which will be described in the following sections.

*4.3. The Post-Production Effect after Adopting GI Guidelines*

The MPIG-K2J vision of exporting high-quality products at a fair price through the GI Koerintji cinnamon certification is in line with TAKTIK. The growing concern about food quality and safety was MPIG-K2J's main highlight. Regarding the provision of quality and safety, a relative improvement practice can be found in Kerinci to add quality and safety to place-based food production. Furthermore, consumers demanding more safe food ascribe growing importance to food products' origin. Here, MPIG-K2J, as the initiator of GI Koerintji cinnamon, had several responsibilities in the cinnamon farmers' community, including: (i) to maintain the quality of goods produced following GIs criteria in the guidebook; (ii) to communicate the quality and characteristics of specific GIs products to consumers; and (iii) to recommend sales prices for GIs goods based on their characteristics and uniqueness. To fulfill its compliance, MPIG-K2J works with the handbook of requirements for GI Koerintji cinnamon to guide TAKTIK. The book contains information on the description concerning the quality and specific characteristics of a good. This information can be used to differentiate goods of the same category. The book of requirements highlights five significant aspects relevant to research by Munsugu et al. [21].

The GI's handbook standard specifically demands technical upgrades to conform to the recognized code of practice (for example, the sun-drying and processing process, as shown in Table 3. As a result, following the criteria and standards can impact increased value and profitability for the future.

*4.4. Validating the Impact of GI Guidelines*

The research continues to survey the members of TAKTIK whether they adopt the guidelines that MPIG-K2J previously established. Production centers were examined to compare and predict the outcome for DM and PM. All the processes were documented to

determine whether the guideline helped harvesters improve practices that can be seen in Figure 7, where members of the group used mat to sundry.

**Table 3.** Drying and processing methods for cinnamon based on the handbook of requirements for the GI for Koerintji Cinnamon, p. 12.

| No. | Drying Methods (DM.) | Processing Methods (PM) |
|---|---|---|
| 1 | Sun drying is for cinnamon bark, differentiated into KM, KF, KS, KA grade, and sticks and powder. | Cinnamon bark that has been scrubbed and dried to 10–14% of moisture content is separated according to gradation |
| 2 | The drying process is carried out using sunlight | Cinnamon bark, which is graded as KM-KF-KS-KA, is separated for further processing into powder or sold directly without using the name Koerintji cinnamon |
| 3 | The drying should be clean, using tarpaulin/plastic as a base for drying or using a table. | Cinnamon bark, separated according to gradation, is then cut into according to buyer preferences. |
| 4 | Cinnamon drying must not be located on the side of the public road and streets | Packing is done using a gunny sack. |
| 5 | Drying must be done for 2–3 days, then dried for seven days in an airtight room with a temperature of 26 degrees Celsius | The stitching of the gunny sack may not use ropes lubricated with kerosene, engine oil, diesel oil. However, the use of water to lubricate the gunny sack is permitted. |
| 6 | The drying process is finished when it becomes brown-reddish with a 10–14% moisture content. A harvester can recognize it if the cinnamon bark is easily broken. | The size of the net or cardboard used is adjusted to market demand. The gunny sack and cardboard are labeled Koerintji cinnamon Geographical Indications and ready to be marketed. |

Note: Members were interviewed voluntarily during the assessment. Source: Research investigation in Talang Kemuning, 2017.

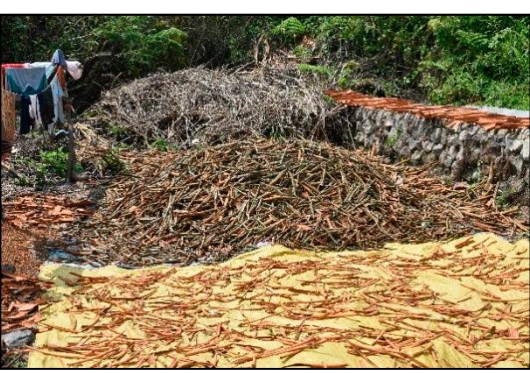

**Figure 7.** TAKTIK members used the mat to sundry (coordinate: −2°14.232′, 101°30.735′). Source: documentation of researcher.

During the observation, attempts were made to validate the points of 1–6 processes. The research areas were clustered with the distance from TAKTIK coordination offices with an area A (1–3 km above), B (3–6 km above), and C (6 km above).

According to the book of requirements (Table 4), the same respondents who implemented DM and PM were interviewed.

**Table 4.** The surveyed areas that had implemented the book of requirements guidelines.

| AREA. | Sundried Two-Three Days | | Using Tarpaulin/Plastic | | Checking the Moisture Content | | Sun-Drying on Safe-Place | | Safely Save the Commodity Indoor | | Separate by Grade | | Check for Foreign Objects | |
|---|---|---|---|---|---|---|---|---|---|---|---|---|---|---|
| | Yes | No | Yes | No | Yes | No | Yes | No | Yes | No | Yes | No | Yes | No |
| A (*n* = 20) | 13 | 7 | 18 | 2 | 15 | 5 | 17 | 3 | 14 | 6 | 18 | 2 | 16 | 4 |
| B (*n* = 20) | 12 | 8 | 19 | 1 | 11 | 9 | 14 | 6 | 12 | 8 | 17 | 3 | 13 | 7 |
| C (*n* = 20) | 9 | 11 | 15 | 5 | 10 | 10 | 12 | 8 | 11 | 9 | 12 | 8 | 11 | 9 |

Based on this table, the distance between areas could affect the closeness to the TAK-TIK, which determines the practices for following the guidelines. TAKTIK continuously reminds its members of the use of GIs guidelines in the process. However, the purposes and benefits of the implication of GIs are not widely known or understood by cinnamon farmers. In this surveyed area, A, B, and C (*n* = 60) (mostly from area C) did not make sundried cinnamon in a safe place (*n* = 8). Furthermore, they did not check for foreign objects hazardous to the human body (*n* = 9). Most farmers who implemented the DM and PM based on GAP and GMP adjustments live in area A (*n* = 20).

### 4.5. The Effect of Internal Control System

One development is through the Internal Controlling System (ICS) that adds value to DM and PM. An ICS is an element of a recorded quality assurance system that allows an external certification body to delegate to an established body or unit within the certified operator the periodic inspection of individual group members [49]. It ensures that third-party certification bodies, such as MPIG-K2J, conduct an audit of the system's proper functioning and carry out a few spot-check re-inspections of TAKTIK. Quality assurance helps safeguard and develop TAKTIKs activities (Figure 8). Therefore, the ICS's presence may account for the organization's specific requirements where it operates, adds real value, and helps manage farmers' groups pragmatically. In this survey, "value-added" means the quantified value of an investment created by supporting agencies' willingness to develop a functioning ICS for TAKTIK.

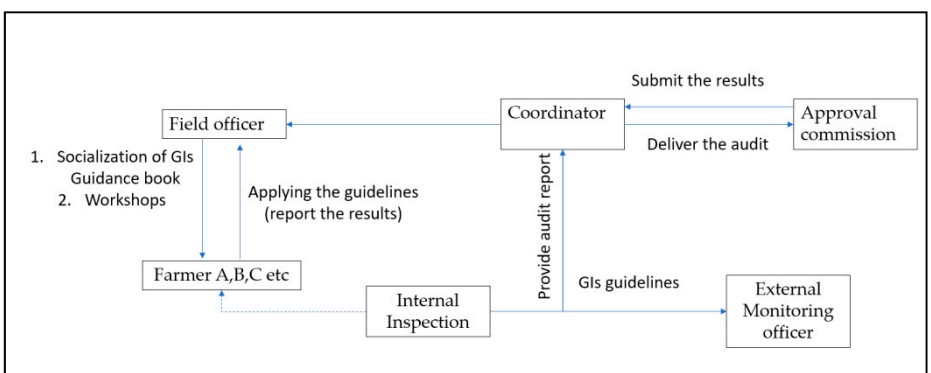

**Figure 8.** Internal control system adopted from GIs handbook of requirements. Source: TAKTIK [consent for publication].

In analyzing the presence of ICS in the GI Koerintji cinnamon as a value chain improvement, we refer to the concept of Porter [41], namely the main activity and the supporting activities. According to the MPIG chairman, ICS requires farmers to implement self-control based on the operating manual and control plan. Furthermore, external controls by MPIG-K2J on producers' compliance with the control manual and plan are executed at least once a year. The quality of the product will be monitored for its compliance.

The controlling system determines the responsibilities, identifies crucial points of control, specifies the corresponding inspection methods, and stipulates sanctions. Therefore, the operating manual and the control plan need to comply to the handbook. The ICS developed from the handbook of requirements before and after production is described in the Figure 9 below.

Figure 9 shows that ICS in GIs brought an added value in the harvesting and post-harvesting processes. The partnership between TAKTIK, Rikolto, and MPIG-K2J was quite adequate because they had created an ICS based on ATN, which requested the farmers' group to ensure quality control and product traceability.

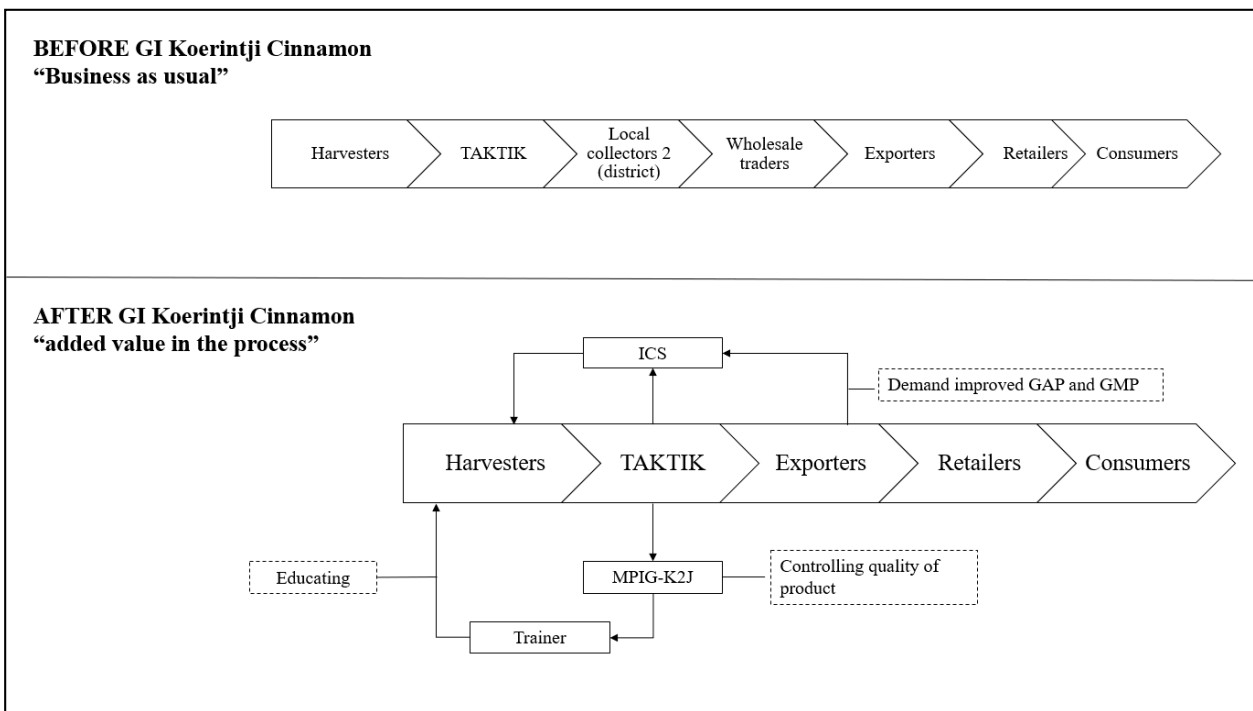

**Figure 9.** Value chain before and after GI Koerintji cinnamon.

Once every three months, MPIG-K2J inspects and audits them. In 2017, the registered members of TAKTIK were 254. However, 136 adopted the ICS system based on the GI Koerintji cinnamon guideline book, which means it was a challenging job for MPIG-K2J. They conducted a thrice-monthly visitation to the TAKTIK members to monitor and evaluate their post-harvesting practice, including sun-drying and packaging, since December 2016.

Since adopting and applying the I.C.S., TAKTIKs' members in the post-harvesting handling and process have become more thorough on foreign objects. Furthermore, the quality control by TAKTIK has become stricter than before. Hence, the quality of cinnamon is guaranteed. Furthermore, TAKTIK activities in coordination with their members were examined, including whether the ICS follows the GI Koerintji cinnamon guideline. Finally, the survey investigated how the practice was conducted and their active members and determined improved DM and PM methods.

The TAKTIKs' product handling was investigated, as shown in Figure 10. The investigation and observation is to see movement and handling of materials and products in a systematic manner from point of origin to warehouse. The main objective is tKerino observe the safety of the products and how workers conduct proper commodity handling techniques of the cinnamon.

The process involved: (1) Using a mat for the cinnamon bark's sun-drying process after being brought from the forest (the sun-drying process can take up to three days during the daytime, from 9 AM—4 PM, to decrease the moisture content up to 15%); (2) separating the grades (KM/KF/KS/KA); (3) after the sortation, the cinnamon categorized as KA is then cut into the size of 5, 8, and 12 cm long, depending on the buyers' requirements; (4) before putting the cinnamon in the nets and cardboard, foreign objects in the cinnamon, such as rocks, rubbers, and plastic, were checked; (5 and 6) After selection, TAKTIK wraps the cinnamon in a nest or puts it in cardboard based on the buyer's demand. Finally, the bag is put, and a sign of product origin is explained in the following section.

The improved post-harvest handling, such as drying and processing-packaging, are farmers' behavioral changes. They became fully aware of the hazardous impact of a foreign object in cinnamon bark. In Figure 11 shows the function of GIs among TAKTIK members. The total respondent (*n* = 20) included all the TAKTIK members' living in area A. The

four most crucial question-related GIs and their added value for their daily production were examined.

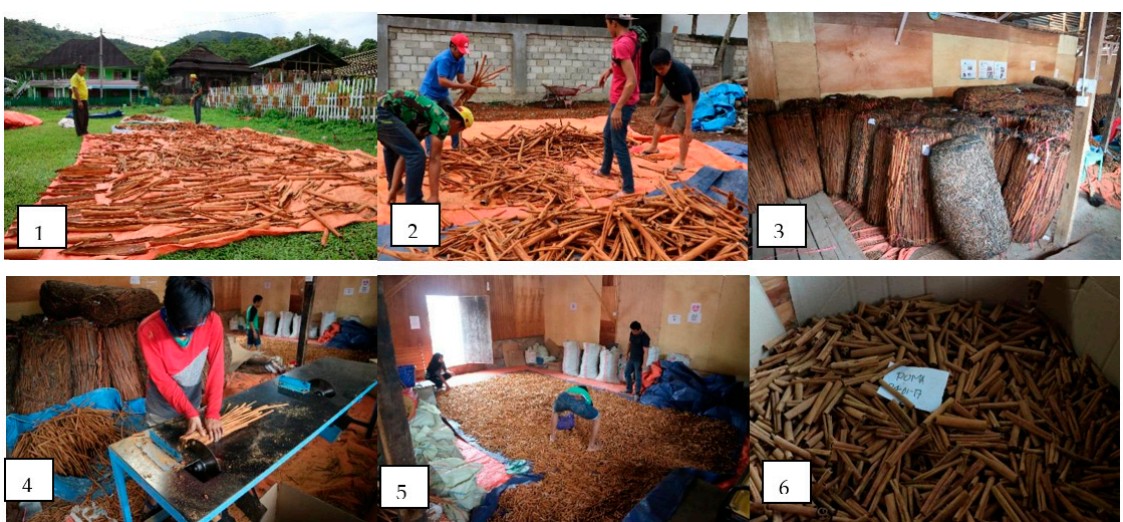

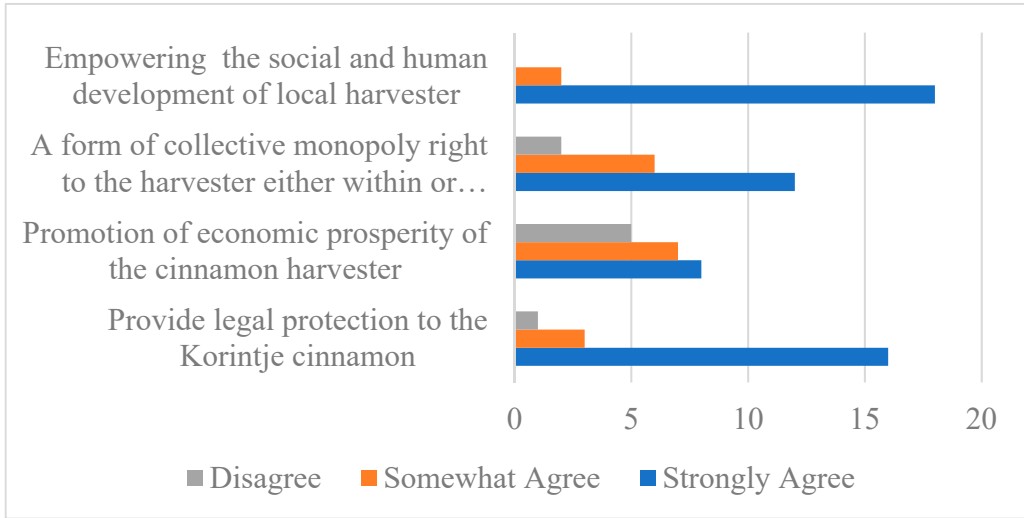

**Figure 10.** TAKTIK post-harvesting practice following the GI Koerintji handbook guideline.

**Figure 11.** The function of GI for TAKTIK members.

The respondents responded positively to GI Koerintji cinnamon's presence, though not all of them clearly understand how it may boost their livelihood and improve post-harvesting practice. Therefore, Koerintji cinnamon's guideline changed their traditional cultivation and harvesting practice, with eighteen ($n = 18$) respondents agreeing to it. The interesting part was the question concerning whether GI brings economic prosperity for the farmers. The female members strongly agrees ($n = 7$), but the male ($n = 5$) disagree and some of them are in between ($n = 6$)

However, the majority of the members ($n = 16$) consent that the GI also became legal protection as Koerintji cinnamon originality due to the fact that while other regions also produced cinnamon, it is not as high premium a commodity as that from Kerinci regency. Due to its functionality, GIs are "place-based names that conveyed the geographical origin, as well as the cultural and historical identity of agricultural products" [50]. In this case, it also brings opportunity for the Koerintji cinnamon. Revinon [51] showed that using the GI label on the local market also has a brand awareness impact for the region or a group of villages, leading to economic improvement. International agreements show that GI is a certification that "identifies a good as originating in a delimited territory or region where a

noted quality, reputation or another characteristic of the good is essentially attributable to its geographical origin and the human or natural factors there" [50]. This is particularly interesting in the Koerintji cinnamon case, wherein the stakeholder with interest tries to sell the product in the global market. Indonesian cinnamon has an outstanding reputation in the European markets, though the name used in these markets is "Cassiavera". It is produced and originates mainly from the Kerinci regency.

### 4.6. The Effect of Adopting GI Principles for Higher Farmgate Price

Within a year, TAKTIK improved their production, reducing the moisture content up to 25% by adopting the ICS system (one of the handbooks of requirements) including peeling, cleaning, and sun drying. Besides it, the development of ICS creation of a traceability system and putting GI's label in their packaging increased farm-gate price by 4%. In addition, MPIG-K2J conducted a lab analysis for volatile oil (VO) containments to verify the cinnamon quality in order to meet the buyer's expectations related to the farmgate price.

One reason for this is that the area of production of TAKTIK was not located as one of the favorites and the highest concentration of volatile oil in the area, as stated in the handbook of GI Korintjie cinnamon (see Table 5).

**Table 5.** Chemical contents of Korintjie cinnamon produced in six sub-districts.

| No | Sub-District | Volatile Oil (%) | Cinnamaldehyde (%) |
|----|--------------|------------------|--------------------|
| 1 | Gunung Raya | 5.80 | 93.23 |
| 2 | Bukit Kerman | 2 | 90.50 |
| 3 | Batang Merangin | 1.78 | 90.08 |
| 4 | Gunung Kerinci | 3.57 | 86.04 |
| 5 | Siulak Deras | 2.20 | 74.35 |
| 6 | Siulak Mukai | 2.83 | 73.48 |

Source: Handbook of requirements, p. 15.

Cinnamon from these sub-districts was tested through laboratory assessment. The table shows that TAKTIK operational activities are in the Bukit Kerman sub-district that produces 2% of volatile oil. Its fixed buyer contract agreement from PT Agripro Tridaya Nusantara (ATN) only tolerates water contaminants up to 20–25% when the VO percentage is above 3.5%. ATN offered a price of €2.92–3.11 for one kilogram of KM with this quality. This led to a winning stage for TAKTIK, where they had to meet all the ATN criteria. Table 6 shows the sale price difference of TAKTIK over time.

**Table 6.** Price differentiated over time (improved).

| Period | Grade | Moisture Content (MC) | VO | Farm-Gate Price |
|--------|-------|-----------------------|----|-----------------|
| June 2016 | KM | 35–40% | 2.7% | 2.34 |
| | KF | | | 2.16 |
| | KS | | | 1.98 |
| | KA | | | 1.86 |
| June 2017 | KM | 23–25% | 4% | 2.92 |
| | KF | | | 2.79 |
| | KS | | | 2.66 |
| | KA | | | 2.53 |

Source: TAKTIK yearly progress report and documentation, 2018 [consent for publication].

TAKTIK had gained confidence because of demonstrating its influence in setting the price. MPIG-K2J became involved in training, assisting the producers in applying ICS to reduce the moisture content to 25 percent and forbid pesticide and chemical fertilizer use.

In return, buyers witnessed first-hand that producers' capability in producing premium quality products is improving. TAKTIK, with the support of the Jambi provincial agriculture agency, tested its volatile oil percentage in the laboratory of the Indonesian Center for Estate Crops Research and Development in May 2017.

Buyers also give regular feedbacks to encourage further improvement. As the mutual trust between the two parties strengthens, the commitment to a long-term partnership/cooperation can be realized. The partnership between TAKTIK and ATN became advanced and developed into a joint operation to pursue several goals, including accessing potential markets, gaining efficiencies, obtaining a loan for a significant investment (in this case, a warehouse), and access to skills and capabilities development.

Additionally, the joint operation between TAKTIK and ATN, including ATN's end buyer (LP), could assure financial institutions (banks) hoping to access individual loans for TAKTIK members' capitals in harvesting seasons. This is evident for the trust given of mutual benefit, where they improve their practice within one year. Therefore, the impact of geographic indications on farmers' income could be increased over time when they also enhance their approach to meet the buyers' criteria for a high premium commodity (Figure 12).

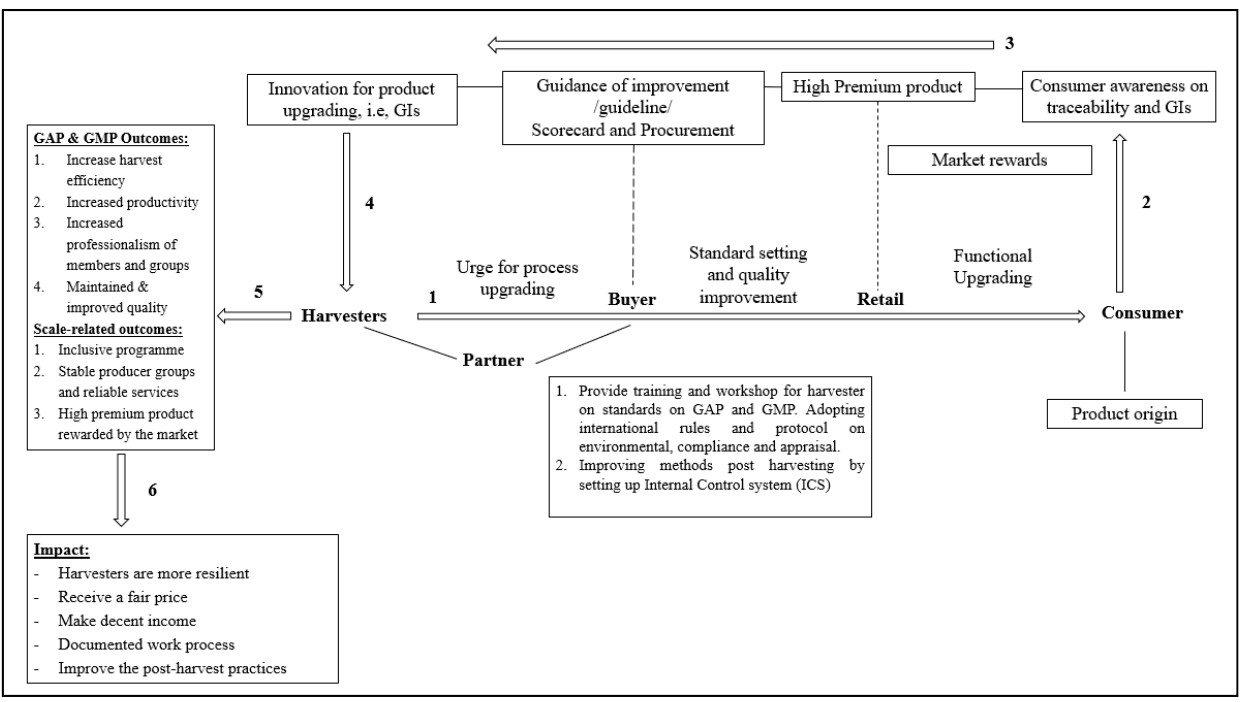

**Figure 12.** The GIs Koerintji cinnamon is an added value to meet buyers' criteria.

Figure 12 shows the flowchart of the added value, in which the subject concern from ATN and LP producers, in this case, was that TAKTIK should have met the good agriculture practices (GAP) in processing the goods, and also good manufacturing practices (GMP), as safety and wholesomeness are of utmost concern to the spice industry, customers, consumers, and regulators. As any other agricultural product, the safety, quality, and consistency of cinnamon may be compromised during the processes which occur between the farm and fork.

### 4.7. The Effect of the Label for Its Origin

The branding of origin represents one of the possible strategies to escape the common goods market. Therefore, at low prices and intense competition [32], the branding of Koerintji cinnamon with GI labels promotes the brand in the global market. The importance of differentiating branding and promoting Koerintji cinnamon as a worldwide brand in

target markets is paramount to highlight the main characteristics of Indonesian cinnamon sourced from other areas for its comparative advantage. The Koerintji's cinnamon brand image includes organic produce, high volatile oil, unique taste, and a peppery, robust flavor. It could have several positive effects when product originality is stated, which appears to be a good marketing strategy [52]. Providing a specific label for cinnamon is punctual in spice trading, identifying its locality and traceability. TAKTIK also uses the label for its branding of origin at its packaging. According to handbook principles, the label reduces misappropriations and encourages other cinnamon harvesters in the regency to adopt GI standards (Figure 13).

Use of GI label in TAKTIK processed cinnamon powder packaging

**Figure 13.** TAKTIK powder cinnamon packaging with GI Koerintji label.

Using the label may encourage consumers to pay more for the certified product and know its origin [53]. Establishing an adequately protected cinnamon origin (*seu generis*) brand can help harvester gain a competitive advantage in buyer-driven global markets. Adding value by differentiating the product increases consumer perception of a premium commodity. Kampft research shows that consumers are willing to buy at a price 10% higher for products protected by GIs, with around 87% of them ready to pay 20% more [54]. Furthermore, consumers' information of origin can be appreciated, differentiating them from farmers' output in the rest of the world (see Figure 14) [11].

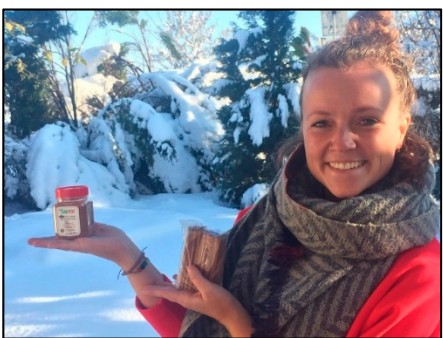

**Figure 14.** Cinnamon packaging with a GI label.

Geographical indication protection should ideally provide consumers with information and assurance that protection for specific products is provided based on their uniqueness, implying that they are unavailable in other regions and thus satisfying consumers when purchasing or consuming such products [41].

## 5. Conclusions

This research has shown indications of the increase in the farmers' income in the GI period. The premium product demanded of ATN meets the criteria of newly improved techniques that harvesters' farmers adopt due to the necessity of the GI guidance book provided by MPIG-K2J. Furthermore, a GIs certificate guarantees the correctness of the quality and origin of cinnamon Koerintji in the market.

The price increase was expected to positively affect cinnamon products' development and production and increase the Koerintji cinnamon sell price. As place-based name protection, GI Koerintji cinnamon is an added value due to several factors, including (1) clear product identification; (2) avoiding fraudulent competitive practices and providing protection to consumers from misuse of the reputation of GIs; (3) quality guarantee which gives confidence to consumers; and (4) fostering local producers, supporting coordination, and strengthening peer rights-holder organizations to create, provide and strengthen product name and reputation image. This, in turn, increases the value derived from their products while at the same time providing consumers with information needed that impacts prices as well. According to Daphne Zografos, GIs may increase production output and value-added products, including superior agricultural products. The added value is shown by the increase in the price of geographical indication products in the market, which is more expensive than similar products without GIs [41]. Additional benefits include: (i) GI can be used as a product marketing strategy at local and global market levels; (ii) GIs adds value to the product and improves the harvesting activities; (iii) products with GIs label can be traceable using a tracking code from its origin to avoid fraudulent competitions; and (iv) GIs are leveraged to gain extra gate price from a buyer.

Although the GI of Koerintji cinnamon impacts the increasing sale price and label of origin, it has been demonstrated that GI development could also directly enhance biodiversity conservation through the utilization of a specific genetic resource or indirectly through production and management strategies that incorporate landscape and ecosystem factors. The fact that governance and market success contribute to the viability of rural livelihoods that rely on the sustainable use of certain biological and genetic resources has direct benefits in terms of sustainability in rural landscapes.

They usually only bring environmental benefits if ecological standards are stipulated in the code of practices. This unchartered area can be motivation for further research and study.

**Author Contributions:** Several researchers have contributed to this research since its actualization. Conceptualization S.R.M.; methodology S.R.M. and W.V.; software S.R.M.; validation S.R.M. and D.R.A.M.; formal analysis S.R.M.; investigation S.R.M. and A.R.; resources S.R.M., A.R. and D.R.A.M.; data curation D.R.A.M.; writing—original draft preparation S.R.M.; writing—review and editing W.V. and S.S.; visualization S.R.M.; supervision P.V.D. and S.S.; project administration P.V.D. All authors have read and agreed to the published version of the manuscript.

**Funding:** This research is funded by the Indonesia Endowment Fund for Education (LPDP).

**Informed Consent Statement:** Informed consent was obtained from all subjects involved in the study.

**Data Availability Statement:** Data sharing not applicable.

**Acknowledgments:** The research project on which this paper is based was made possible by an education fund from the Indonesia Endowment Fund for Education (LPDP).

**Conflicts of Interest:** The authors declare no conflict of interest.

**Appendix A**

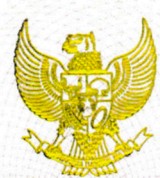

REPUBLIK INDONESIA
KEMENTERIAN HUKUM DAN HAK ASASI MANUSIA

**SERTIFIKAT INDIKASI-GEOGRAFIS**

Menteri Hukum dan Hak Asasi Manusia atas nama Negara Republik Indonesia berdasarkan Undang-Undang Nomor 15 Tahun 2001 tentang Merek jo Peraturan Pemerintah No. 51 Tahun 2007 tentang Indikasi-Geografis, memberikan Hak Indikasi-Geografis kepada :

Nama dan Alamat Pemilik Indikasi-Geografis
: Masyarakat Perlindungan Indikasi Geografis Kayumanis Koerintji Jambi (MPIG-K2J) Jl. Simpang Empat, Desa Lempur, Kecamatan Gunung Raya, Kabupaten Kerinci Provinsi Jambi

Untuk Indikasi-Geografis dengan Nama
: KAYUMANIS KOERINTJI

Nama Produk
: Kayumanis

Tanggal Penerimaan
: 23 Desember 2015

Nomor Pendaftaran
: ID G 000000043

Tanggal Pendaftaran
: 26 Mei 2016

Perlindungan Hak Indikasi-Geografis tersebut diberikan selama karakteristik khas dan kualitas yang menjadi dasar bagi perlindungan atas Indikasi-Geografis tersebut masih ada. Sertifikat Indikasi-Geografis dilampiri dengan buku persyaratan yang tidak terpisahkan dari sertifikat ini.

a.n. MENTERI HUKUM DAN HAK ASASI MANUSIA
REPUBLIK INDONESIA
DIREKTUR JENDERAL KEKAYAAN INTELEKTUAL

Prof. Dr. Ahmad M. Ramli,SH.,MH.,FCBArb.
NIP. 196107041987011001

**Figure 1.** Certificate of Geographical Indication. Source: Handbook of Requirements, MPIG-K2J [Consent for publication].

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
