# Peer review of "The Effect of Geographical Indications (GIs) on the Koerintji Cinnamon Sales Price and Information of Origin"

_agronomy, doi:10.3390/agronomy11071410_

Round 1
Reviewer 1 Report
The submitted manuscript is thematically suitable to "Agronomy" journal. It provides a detailed account of the introduction of Geographical Indication to cinnamon production in Korintji, Indonesia.
I would like to make several general points that should be addressed in my opinion before the publication of this text:
- Research aim is stated several times in the manuscript, each time in a different form. Authors must clearly state the research aim and be consistent throughout the text.
- The manuscript reads more like a story. Scientific added value of the text should be properly presented.
- Hypothesis (or hypotheses) must be clearly stated. The way how they were verified needs to be well described.
- Statistical representativity of the studied samples of respondents must be discussed.
- Limitations of the study need to be stated and discussed as well as possible future research directions.
- Size and structure of the manuscript should be revised to focus on the scientific method rather that journalistic account.
Author Response
Dear reviewer
Thank you a lot for your patience. I am sorry for the long reply, my child was sick and needed special treatment.

Reviewer 2 Report
The topic addressed in this paper is interesting as a case study addressed to the Korintje cinnamon (geographical Indication), but I think that it should be improved from a methodological point of view. I In fact, there are some lacks from the theoretical side. The survey carried out using a semi-structured questionnaire, roundtable discussions and interviews provides qualitative and only descriptive results. I believe it is necessary to undertake a set of modifications, in order to improve the current manuscript, to fulfil the quality standards of the journal. I think it can be published, after major revisions.
- I would revise the Introduction which is mainly focused on the product characteristics rather than presenting the field of research. I would write more background about the research topic, including recent literature in the field and expand citations from other current papers that reviews the literature (especially as the value chain assessment).
- Several methods and techniques could be adopted to manage workshop with stakeholders (Multi-Actor Approach), it is necessary to expand the working hypotheses, specify research problems and providing how they have been addressed.
- More information is required about the method, the authors need to include more theoretical underpinnings to their work and then draw out the implications of the results.
- As the premium price and the willingness to pay a higher price when the product is certified need more comparison with other studies (other similar products).
- Because of the lack of a proper current literature review the results are not discussed in the context of previous findings.
Author Response

(The authors gave the same response as above.)

Reviewer 3 Report
Dear Authors,
The article is very interesting but it requires to be properly organized. Do not place methodology in the Result section. Provide the necessary information in the Methodology section (Lns 393-403)
The research goal should be at the end of the introduction (Lns 82-88). Section 3.1 is a literature review, it does not refer to the results of the authors' research, this content should be included in the Introduction
Figure 3. Stakeholder perception for GIs during the round table discussion. Wrong Figure numeration since Figure 5. Picture 3 should be 5.
I think photographs with people are not necessary in this study.
Lack of description Figure 3 and 10, no units of measure.
Ln 273: It is not Fig.8 (in the text)
Lack of referring to Figures in text (Fig. 3,
Ln 367: Figure 10 shows the whole value chain process. It is not Fig. 10
Lns 380-382: The text does not refer to figure 5.
Table 2 and description should be in section methodology.
Discussion needs to be improved as well. It must make clear what is completely new in the presented results.
Lns 607-611: not quoted in the summary (Daphne Zografos). This sentence should be included in the discussion section or Introduction section
Recerences are not provided in accordance with the requirements of the Journal. The reference style seems to be different from the one for Agronomy
Author Response

(The authors gave the same response as above.)
